# Health care workers' self-perceived infection risk and COVID-19 vaccine uptake: A mixed methods study

Kasusu Klint Nyamuryekung'e[1]*, Maryam Amour[2], Innocent Mboya[3,4], Harrieth Ndumwa[2], James Kengia[5], Belinda J. Njiro[2], Lwidiko Mhamilawa[6,7], Elizabeth Shayo[8], Frida Ngalesoni[9], Ntuli Kapologwe[5], Albino Kalolo[10], Emmy Metta[11], Sia Msuya[3,4,12]

1 Department of Community Dentistry, School of Dentistry, Muhimbili University of Health and Allied Sciences, Dar es Salaam, Tanzania, 2 Department of Community Health, School of Public Health and Social Sciences, Muhimbili University of Health and Allied Sciences, Dar es Salaam, Tanzania, 3 Department of Epidemiology and Biostatistics, Institute of Public Health, Kilimanjaro Christian Medical University College, Moshi, Tanzania, 4 Community Health Department, Institute of Public Health, Kilimanjaro Christian Medical University College, Moshi, Tanzania, 5 Presidents Office Regional Administration and Local Government, Dodoma, Tanzania, 6 Department of Parasitology and Medical Entomology, Muhimbili University of Health and Allied Sciences, Dar es Salaam, Tanzania, 7 Department of Women's and Children's Health, International Maternal and Child Health (IMCH), Uppsala University, Uppsala, Sweden, 8 National Institute for Medical Research, Dar es Salaam, Tanzania, 9 AMREF Health Africa in Tanzania, Dar es Salaam, Tanzania, 10 Department of Public Health, St. Francis University College of Health and Allied Sciences, Morogoro, Tanzania, 11 Department of Behavioral Sciences, School of Public Health and Social Sciences, Muhimbili University of Health and Allied Sciences, Dar es Salaam, Tanzania, 12 Department of Community Medicine, Kilimanjaro Christian Medical Centre, Moshi, Tanzania

* kasusuklint@yahoo.com

**Data Availability Statement:** All data and related metadata underlying the findings reported in a submitted manuscript should be deposited in an appropriate public repository, Dryad with the

## Abstract

Vaccination is the most cost-effective way of preventing Coronavirus Disease 2019 (COVID-19) although there was a considerable delay in its institution in Tanzania. This study assessed health care workers' (HCWs) self-perceived infection risk and uptake of COVID-19 vaccines. A concurrent embedded, mixed methods design was utilized to collect data among HCWs in seven Tanzanian regions. Quantitative data was collected using a validated, pre-piloted, interviewer administered questionnaire whereas in-depth interviews (IDIs) and focus group discussions (FGDs) gathered qualitative data. Descriptive analyses were performed while chi-square test and logistic regression were used to test for associations across categories. Thematic analysis was used to analyze the qualitative data. A total of 1,368 HCWs responded to the quantitative tool, 26 participated in the IDIs and 74 in FGDs. About half of the HCW (53.6%) reported to have been vaccinated and three quarters (75.5%) self-perceived to be at a high risk of acquiring COVID-19 infection. High perceived infection risk was associated with increased COVID-19 vaccine uptake (OR 1.535). Participants perceived that the nature of their work and the working environment in the health facilities increased their infection risk. Limited availability and use of personal protective equipment (PPE) was reported to elevate the perceived infection risks. Participants in the oldest age group and from low and mid-level health care facilities had higher proportions with a high-risk perception of acquiring COVID-19 infection. Only about half of the HCWs

following DOI details: https://doi.org/10.5061/dryad.vdncjsxz9

**Funding:** This research work was supported by grant from UNICEF (MA). The funders had no role in study design, data collection and analysis, decision to publish, or preparation of the manuscript.

**Competing interests:** The authors have declared that no competing interests exist.

reported to be vaccinated albeit the majority recounted higher perception of risk to contracting COVID-19 due to their working environment, including limited availability and use of PPE. Efforts to address heightened perceived-risks should include improving the working environment, availability of PPE and continue updating HCWs on the benefits of COVID-19 vaccine to limit their infection risks and consequent transmission to their patients and public.

## Introduction

The novel coronavirus disease 2019 (COVID-19) caused by severe acute respiratory syndrome corona virus 2 (SARs-CoV-2) remains a significant disease of public health concern. Since its emergence, it has been shown to spread rapidly causing dramatic global health crisis [1]. It was declared as a pandemic by the World Health Organization (WHO) on the 11[th] of March 2020 as of April 26[th], 2023 there was a total of 764,474,387 confirmed cases and 6,915,286 confirmed deaths globally [2]. Tanzania has reported relatively few number of COVID-19 cases, with a total of 42,973 confirmed cases and 846 deaths reported between January 2020 and April 26, 2023 [3].

Vaccination is among the most cost-effective ways of preventing diseases. For the vaccination effect to be appreciated, several strategies to be considered include vaccine availability, accessibility, and acceptability of the population to vaccinate. Studies show that about 14.3% and 22.1% of the global population intend to refuse vaccination or showed uncertainty respectively, with higher rates reported in lower income countries [4]. Moreover, perceived vaccine efficacy and safety concerns contribute to the observed trends in most countries [4].

There was a considerable delay in COVID-19 vaccine roll out in Tanzania, after launching the first nation-wide COVID-19 vaccination, the first roll out was made available among priority groups; health care workers (HCWs) with high risk of getting and transmitting the infection, people with advanced age and underlying medical conditions with a high risk of developing severe disease [5]. In Tanzania, a total of 39,392,419 vaccine doses were administered by the 22[nd] of March, 2023 under COVAX facility [3, 6]. Currently, access to COVID-19 vaccine in Africa has improved; however, vaccines acceptance and hesitancy influenced by social, political, and religious factors may contribute to low vaccine uptake [7, 8].

HCWs stand as among the most important groups as trusted influencers in regards to health issues, including vaccination decisions [7]. This important group should be guided and supported to provide credible and scientifically proven information on vaccines as their influence in the community remains pivotal. It is therefore important to understand and acknowledge HCWs perspectives with regard to COVID-19 vaccines [4]. However, studies have reported a number of challenges facing this population including high risk of infection, insufficient personal protective equipment (PPE), heavy workloads and discrimination [8]. In Tanzania, a national COVID-19 committee was formed in May 2021, which recommended that HCWs be vaccinated as a priority group.

Risk perception, defined as an individual perceived susceptibility to threat, plays a key role in health behavioral change theories, including health decision making process [9]. HCWs are among the most vulnerable groups for SARs-CoV-2 infection, they work in frontline positions with suspected and confirmed COVID-19 cases, 5–7.3% of HCWs were found to be COVID-19 positive in some developed countries [10, 11]. Some studies have reported that perceived risk of COVID-19 infection and detrimental health effects among HCW are associated positive protective behaviors [12]. In Ethiopia, 88% of HCWs were reported to perceive their risk of

being infected with COVID-19 infection as high, and showed widespread practice on preventive measures [13].

Myths and misconceptions around the COVID-19 vaccine subject have been circulating and its impact can be observed especially in developing countries [1]. This has been shown to contribute to the observed vaccine hesitancy, which is defined by WHO as "the reluctance in accepting vaccines or an outright refusal of vaccines despite their availability" [7]. WHO has further mentioned vaccine hesitancy as one of the top global threats to public health in 2019 [7, 14]. As reported by a study done in Senegal with 5.5% COVID-19 vaccine coverage, vaccine hesitancy and refusal have contributed to low vaccine uptake despite its multifaceted nature [15]. In Ethiopia, more than 50% of HCWs were found to be vaccine hesitant [4].

While being at an increased risk of COVID-19 infection and disease transmission in Tanzania, HCWs play an influential role in community understanding and overall vaccine uptake; there is paucity of data on the status of vaccine uptake among HCWs and the influencing factors in Tanzania. Understanding HCWs risk perception and their influence on vaccination is crucial in informing policy makers and highlighting educational needs to address the situation especially in developing countries like Tanzania. This mixed method study illustrates on HCWs perceptions in relation to the COVID-19 vaccine uptake situation in Tanzania.

## Materials and methods

### Study setting

Tanzania is a large East African country with an area of 947,000 square kilometers and an estimated population of 61.5 million (2021 World Bank projections) of which about two-thirds live in rural areas. Tanzania comprises of the much larger mainland and semi-autonomous isles (Zanzibar); the current study was conducted in mainland Tanzania. Mainland Tanzania is administratively divided into 26 regions, each region comprising of a variable number of districts (4–6), which in turn contain wards. The Tanzanian healthcare delivery facilities follow this pyramidal administrative arrangement, with regional referral hospitals situated at the apex, functioning as the highest-level hospital within a region, which receive referrals from district hospitals, that in turn receive patients from lower levels (health centers and dispensaries). Groups of regions are further clustered into six geographical zones (Northern, Coastal, Central, Southern highlands, Western and Lake). One region was randomly selected from each of these six zones to be included in the study. The sampled regions were Kilimanjaro, Lindi, Njombe, Mbeya, Tabora and Simiyu representing Northern, Coastal, Central, Southern highlands, Western and Lake zones, respectively. Dar es salaam was purposively selected to be included because it is the largest city in Tanzania, and the major port of entry to the country from international arrivals. The researchers believed that it is important to capture the HCW population perceptions from a cosmopolitan city.

### Study design and participants

A concurrent embedded mixed methods research design was utilized to collect data among HCWs in seven regions of mainland Tanzania from November 2021 to January 2022. The qualitative part of the study was embedded in the quantitative cross-sectional study. The qualitative part was intended to explain the HCWs risk perceptions towards COVID-19 disease, as a supplement of the quantitative study assessing COVID-19 vaccine hesitancy.

Study population from which the sample size was determined included all cadres of HCWs at all levels of healthcare service provision in Tanzania. This included any medical, dental, laboratory, pharmaceutical or nursing staff working in a medical institution as officially registered through their professional associations. A sample size for the quantitative part of the study

(N = 1400) was determined by using a single proportion formula taking a standard normal value of 1.96 under the 95% confidence limit, 50% proportion of vaccine hesitancy (for maximization of sample size), 3.5% margin of error, 1.5 design effect to address the clustering effect while adjusting for a non-response rate of 20%.

Multi-stage sampling technique was employed to recruit HCWs from the seven (7) selected regions for the quantitative part. One Regional Referral hospital, two (2) district hospitals and two (2) health centers from each of the identified regions were included in this study. Therefore, a total of seven (7) Regional Referral Hospitals, fourteen (14) district hospitals, and fourteen (14) health centers were included. Systematic sampling technique was used to select healthcare facilities for inclusion. Sampling of HCWs within the selected health facilities was based on their number in the selected health facilities in a region proportional to their size. Upon determination of the respective health facilities' sample sizes, HCWs were consecutively invited to participate and enrolled into the study. All available HCWs were invited to participate, and interviews carried out until the minimum sample size was reached.

The qualitative component of the study was conducted in four of the seven regions where the quantitative study took place due to resource limitations. In-depth interviews (IDI) were conducted in each region with the key officials including the Regional Medical Officers, Regional Vaccination Officers, District Medical officers, District vaccination officers and Medical Officers In-charge of health facilities leading to a total of 26 interviews. Additionally, we conducted two focus group discussions (FGDs) in each region with participants ranging from 6–12 people leading to a total 74 participants in 8 FGDs. The FGDs engaged HCWs in the selected districts within the study regions.

## Data collection tools and procedures

Quantitative data was collected using a validated, pre-piloted questionnaire through the Open Data Kit (ODK). The pilot testing of the study tools was done through a randomly selected group of HCWs from Muhimbili University of Health and Allied Sciences, to whom the link to the ODK tool was shared for testing. Feedback on the shortcomings of the tools was incorporated in the revised final version. The questionnaires were developed based on various studies and WHO proposed questions to assess vaccine hesitancy and acceptability [16–23]. The questionnaire was prepared in English, translated in Swahili, and had four components: socio-demographic, awareness and knowledge on COVID-19 vaccines, risk perception towards COVID-19 and COVID-19 vaccination status. The questionnaire contained both closed and open-ended questions for assessment of awareness and knowledge components. Risk perception towards COVID-19 infection was measured on Likert scale. Back translation to English was done to preserve the meaning of the questions. The questionnaire was administered face-face by trained research assistants (RAs). On the day of quantitative data collection, the RAs visited the HCWs, introduced themselves and explained the study purpose. Then, they provided study information to the HCWs and obtained informed consent in a quiet, private place around the health facility. Special emphasis was placed on issues of anonymity and confidentiality, and in assuring the respondents that no personal identifiable information will be collected to encourage truthful responses. Only the consenting individuals were interviewed.

Qualitative data was collected through IDIs and FGDs with purposively selected health officials and HCWs to explore their opinions and risk perceptions towards COVID-19. All interviews were conducted in Swahili and audio recorded with the permission of the study participants. Further, researchers applied the principle of bracketing to ensure that pre-understanding information do not influence the data [24]. Furthermore, for enhancement of reliability, field notes as a reflective diary were maintained.

## Data management and analysis

The collected quantitative data was transferred from the ODK to an excel spreadsheet. Upon completion of data collection, each questionnaire was assessed for its completeness. Data entry, cleaning and coding was done using Microsoft Excel program and exported to Stata software V.16.1 (College Station, Texas). Descriptive analyses were performed for proportions, percentages, means and their corresponding standard deviations.

The primary outcome variable of the study was COVID-19 risk perception which was assessed by asking a question "How do you perceive the level of risk that you have for acquiring COVID-19 infection" with responses along a six-point Likert scale ranging from "Not at all" to "Very high risk". Thereafter, the options "not at all, very low risk, low risk, and medium risk" were categorized into "Low risk" and options "high risk and very high risk" into "High risk" for dichotomization of the provided responses. Vaccination status of the respondents was assessed by asking "Have you been vaccinated against COVID-19" with "Yes/No" responses.

Age was categorized into three options (<30 years, 30–39 years and 40+ years) whereas work experience was dichotomized at its median (<6 years, 6+ years). For categorical variables chi-square test and binary logistic regression were used to assess associations between sociodemographic characteristics and COVID-19 vaccination status to risk perceptions. Statistical significance was defined as a p value of <0.05.

For qualitative data, the audio recorded in-depth interviews and focused group discussions were transcribed verbatim into word file documents where non-verbal cues were also considered. The transcription process started within 24 hours after the conduct of the interview to allow follow-up on issues for more clarity and determination of data saturation in subsequent interviews and discussions. The transcribed transcripts were checked against the audio records by two of the research team members to ensure accuracy and quality of the data generated. Thematic analysis was applied to facilitate immersing into the data methodically and thoroughly to identify themes and patterns for gaining in-depth understanding of participants' opinions, experiences, and risk perceptions towards COVID-19 across the dataset. The analysis process followed the five thematic analysis stages as described by Braun and Clarke, 2014 [25] to establish meaningful patterns in the data: familiarization with the data, generating initial codes, searching for themes among codes, reviewing themes and presenting the results. Through this process three main themes emerged: perceived risk of being infected; risk of infecting others and perceived reduced risk to infection due to use of COVID-19 vaccines. The coding also involved identification of the typical quotes that are used to illustrate the various themes presented in the study.

## Ethical considerations

Ethical approval was obtained from the Research and Publication Committee of the Muhimbili University of Health and Allied Sciences (MUHAS-REC-08-2021-839). Permission to collect data in Regions and Councils was sought from the President's Office Regional Administration and Local Government, Ministry of Health Community, Development, Gender, Elderly and Children (MoHCDGEC), Regional Secretariat (RS) and Local Government Authorities (LGAs). Prior to collection of data, all participants were provided with information on the purpose of the study, voluntary nature of participation, right to withdraw from study at any time without consequence and guaranteed anonymity. Signed, informed consent was obtained from all participants before enrolment into the study.

## Results

A total of 1,368 HCWs were approached and involved in the quantitative part of this study. All the approached participants in health facilities consented to participate. Most of the

respondents were female (60.1%) and had the mean age of 35.7 years (SD 10.1). There was almost an equal representation of participants by regions, except for Dar es Salaam which contributed the largest proportion (26.1%). Most of the respondents were from the district-level facilities (42.1%) and about three quarters (77.5%) worked in Government facilities (Table 1).

Only about one half of the HCW (53.6%) reported to have been vaccinated whereas three quarters (75.5%) self-perceived to have a high risk of acquiring a COVID-19 infection.

Logistic regression was used to analyze the effect of perceived COVID-19 risk on the probability of being vaccinated against COVID-19. It was found that the odds of getting vaccinated increased by 1.535 (95% CI 1.197, 1.967) for the HCWs who perceived their risk of getting COVID-19 infection as high (Table 2).

Information on risk perceptions to COVID-19 infection during the qualitative in-depth interviews and focus group discussions revealed three main themes: risk of being infected, risk of infecting others and reduced risk to infection due to use of COVID-19 vaccines. Participants reported that they are at increased risk of being infected by COVID-19 while making a reference to the nature of their work and their working environment. It was explained that the

**Table 1. Background characteristics of HCWs (N = 1368).**

| Variable | Frequency | Percentage |
|---|---|---|
| Age (years) | | |
| <30 | 470 | 34.4 |
| 30–39 | 483 | 35.3 |
| 40+ | 415 | 30.3 |
| Sex | | |
| Male | 546 | 39.9 |
| Female | 822 | 60.1 |
| Education level | | |
| Primary/Secondary | 36 | 2.6 |
| Certificate | 437 | 31.9 |
| Diploma | 610 | 44.6 |
| Degree/Masters | 285 | 20.8 |
| Work experience (years) | | |
| <6 | 731 | 53.4 |
| 6+ | 637 | 46.6 |
| Region | | |
| Dar es salaam | 357 | 26.1 |
| Kilimanjaro | 187 | 13.7 |
| Lindi | 151 | 11.0 |
| Mbeya | 186 | 13.6 |
| Njombe | 158 | 11.5 |
| Simiyu | 137 | 10.0 |
| Tabora | 192 | 14.0 |
| Health facility level | | |
| Regional Referral Hospital | 378 | 27.6 |
| District Hospital | 576 | 42.1 |
| Health center | 414 | 30.3 |
| Facility ownership | | |
| Government | 1060 | 77.5 |
| CDH/DDH | 148 | 10.8 |
| Private/NGO | 160 | 11.7 |

**Table 2. Univariate binary logistic regression model showing relationship between COVID-19 risk perception and vaccination status.**

| Coefficients | B | Std. Error | Sig | | Exp (B) | 95% CI |
|---|---|---|---|---|---|---|
| COVID-19 Risk | .428 | .127 | | .001 | 1.535 | 1.197, 1.967 |
| Constant | -.180 | .110 | | .101 | .835 | |

inadequate and limited use of personal protective equipment's (PPEs) including standard face masks and sanitizers at the health facilities elevated the perceived risks to the infection. Some of the HCWs mentioned to have attended patients several times without using any PPEs due to their frequent unavailability. Others perceived that the that the risk of being infected with COVID-19 infection was so high because even when the PPEs are available, not all HCWs comply with their use. In one of the districts, when officials explained about the working environment and the risks to being infected by COVID-19 infection said:

> "*the risk for health care workers being infected with COVID-19 is high because of the working environment, people have relaxed, they are no longer taking measures against COVID-19, some do not bother to wear masks, wash hands, keep social distancing . . . even when masks are there they just don't put it on all the time as required, everything about COVID-19 seem to have been paralyzed, no one is either complying or discussing about it, which makes the working environment unsafe (IDI1).*

Working in a health facility setting was the other mentioned reason for higher perceived risk of being infected by COVID-19. Participants voiced concerns that it is the HCWs who take care of the COVID-19 patients, which increases their chance of being infected. It was said that most of the hospitals do not have enough offices or changing rooms for HCWs. This necessitates sharing of available rooms among the HCWs. This also included sharing of the same facilities with HCWs attending patients at the intensive care unit (ICU) or patients with difficulty breathing. Participants detailed that the shared rooms are small and limited in number, which was perceived as heightening their risk to COVID-19 infection. When detailing on this matter a participant during the in-depth interviews voiced that unless compliance to the recommend preventive measures is strongly reinforced, HCWs will continue to be at higher risk of being infected by COVID-19:

> "*You cannot say health care workers are not at risk of COVID-19 as far as they are working in the hospital, they are taking care of the COVID-19 patients, they share small rooms, no dedicated rooms for those attending patients at the intensive care unit or with difficult breathing, sometimes do not have all the required PPEs so the risk is there and if one gets infected it is likely the rest will experience the same unless compliance to recommendation protective measures is high we will continue to be at higher risk of COVID-19*" (IDI3)

Furthermore, concerns regarding risks of the HCWs to infect other people, including their patients was voiced. It was reported that frequently the HCWs would attend patients without having knowledge of their personal or attended patient's COVID-19 infection status. As such, it was noted that as long as they attend patients, transmission of the infection was inevitable. This was elaborated during an in-depth interview by the health facility in-charge as follows:

> "*Transmission of the infection is not avoidable . . . as long as you attend patients, a chance of acquiring or transmitting it to others still viable. First of all, you may not even be aware that you are COVID-19 positive when attending a patient because COVID-19 symptoms resemble*

*that of other diseases like malaria such as feeling fever, joint pains, cough and so on. . .and the patient may have same symptoms and you think it is malaria and not COVID-19*" (IDI4).

On the other hand, perceptions that the risk to COVID-19 infection has been reduced with the use of COVID-19 vaccines were also expressed. When expounding on this, participants compared the perceived risk to the infection during the first wave when people were helpless with the time when vaccines were introduced:

"*The risk to COVID-19 infection was very high like 100% during the first wave because COVID-19 was a new thing and we had no enough knowledge on the precautions to take or what to do, but the risk decreased during the second wave because we had enough knowledge on how this disease is transmitted and how to take precautions and the risk decreased more and more to the point that we are not that much worried because there is the introduction of vaccine which has helped us to build protection*" (FGD 3)

Those respondents belonging to the oldest age group had higher proportions with a high-risk perception for acquiring COVID-19 infection compared to the younger age groups. Similarly, the proportion of respondents reporting to have been vaccinated for COVID-19 was highest within the oldest age group. Risk perception and vaccination status was also shown to vary significantly by region of the respondents. Whereas Njombe, Simiyu and Mbeya had more than 85% of the respondents perceiving their risk as high, those in Dar es Salaam only had about 60% reporting the same. On the other hand, respondents from Simiyu, Lindi and Kilimanjaro had more that 60% reporting to have been vaccinated compared to other regions which consistently had less than 50% reporting the same (Table 3).

Respondents from low and mid-level health care facilities (health centers and district hospitals, respectively) reported much higher risk perceptions compared to those in high level facilities (Regional referral hospitals). Equally, respondents in low level facilities had higher proportions of HCWs reporting to have been vaccinated compared to those in high level facilities.

Respondents working in government facilities had a lower proportion (73.5%) reporting to be at high-risk compared to those working in faith-based organizations, NGOs, and private health facilities. Contrariwise, those respondents from the government facilities had higher proportions reporting to have been vaccinated compared to their counterparts (Table 3).

## Discussion

This study aimed at exploring HCWs perceptions in relation to the COVID-19 vaccine uptake to inform policy makers and highlighting targeted educational needs to address similar situations especially in developing countries like Tanzania. About a quarter of the HCW perceived to have a low risk of acquiring a COVID-19 infection. Furthermore, those with perceived low risk had higher proportions reporting to be unvaccinated for COVID-19.

In the current study, majority of the HCWs perceived risk of contracting COVID-19 to be high, consistent to a recent multi-country study that found equally high levels of COVID-19 risk perception levels in the countries [26]. Due to the nature of their daily work activities and physical proximity to potential COVID-19 cases, it was expected that the vast majority of the HCW in health facilities would consider themselves to be at a heightened risk for contracting the infection. However, consistent availability of appropriate and required PPEs would have contributed towards allaying some of the perceived risks. To control the spread of infection, it is crucial that all HCWs become sensitized to the increased risk that they are subjected to with

**Table 3. Socio-demographic characteristics by high COVID-19 risk perception and reporting being vaccinated for COVID-19.**

| Variable | High Risk Perception | *p-value* | Vaccinated | *p-value* |
|---|---|---|---|---|
| **Age (years)** | | | | |
| <30 | 335 (71.4) | | 187 (39.8) | |
| 30–39 | 350 (72.8) | .000 | 271 (56.1) | .000 |
| 40+ | 344 (83.3) | | 273 (65.8) | |
| **Sex** | | | | |
| Male | 399 (73.5) | .159 | 292 (53.5) | 1.000 |
| Female | 630 (76.8) | | 439 (53.4) | |
| **Education level** | | | | |
| Primary/Secondary | 30 (83.3) | | 20 (55.6) | |
| Certificate | 344 (78.9) | .083 | 213 (48.7) | .121 |
| Diploma | 451 (74.4) | | 337 (55.2) | |
| Degree/Masters | 204 (71.6) | | 161 (56.5) | |
| **Work experience (years)** | | | | |
| <6 | 529 (72.8) | .012 | 319 (43.6) | .000 |
| 6+ | 500 (78.6) | | 412 (64.7) | |
| **Region** | | | | |
| Dar es salaam | 211 (59.3) | | 177 (49.6) | |
| Kilimanjaro | 146 (78.1) | | 114 (61.0) | |
| Lindi | 106 (70.2) | .000 | 93 (61.6) | .001 |
| Mbeya | 157 (85.3) | | 87 (46.8) | |
| Njombe | 142 (90.4) | | 78 (49.4) | |
| Simiyu | 114 (83.8) | | 87 (63.5) | |
| Tabora | 153 (79.7) | | 95 (49.5) | |
| **Health facility level** | | | | |
| Regional Referral Hospital | 262 (69.7) | | 174 (46.0) | |
| District Hospital | 454 (79.1) | .004 | 304 (52.8) | .000 |
| Health center | 313 (75.8) | | 253 (61.1) | |
| **Facility ownership** | | | | |
| Government | 777 (73.5) | | 612 (57.7) | |
| CDH/DDH | 121 (81.8) | .006 | 70 (47.3) | .000 |
| Private/NGO | 1313 (82.9) | | 49 (30.6) | |

respect to COVID-19 infection. This may ensure that necessary precautions and protective measures are adopted by the HCW and respective health facilities to prevent acquisition of infections, but more importantly, that they not become the source of infection to the patients and clients that they encounter regularly. High perceived risk of COVID-19 has largely improved the infection prevention and control behaviors of HCWs as indicated by studies in Egypt and Ethiopia [27] however, the picture was different in Tanzania where even though participants reported high perceived risk consistent use of protective gears was not reinforced, even when they were available.

It has been widely reported that high perceived risk of contracting COVID-19 is a significant predictor of vaccine acceptance [28]. We report HCWs that perceived to have a high risk for COVID-19 infection had a larger proportion reporting to have been vaccinated for COVID-19 compared to their counterpart. This is similar to results of from a review indicating that HCWs faced increased exposure to COVID-19 and hence high levels of morbidity and mortality, thereby heightening their own perception of risk, a key factor in the decision to vaccinate [29], [30]. However, current findings reveal that only about a half of the HCW had been

vaccinated despite high perceived risk and sustained efforts to ensure availability and encourage vaccinations. Thus, the link between the perceived risk of COVID-19 and COVID-19 vaccine uptake was tenuous in this setting, contrary to some similar studies [31]. A study in Malawi and Ethiopia found similar results whereby perceived risk of COVID-19 infection was not associated with motivation to receive the vaccine [32, 33]. One probable explanation for this could be that a considerable proportion of people in Africa including Tanzania, consider the vaccines as unnecessary, and that alternatives to COVID-19 vaccination exist. The contextual differences in terms of political, social and cultural factors may explain such dissimilar findings [34, 35]. Many studies have indicated that when people perceive COVID-19 as a threatening disease, the demand for a vaccine against the disease would correspondingly increase. However, this study has shown this to not necessarily be the case. Other factors, especially vaccine safety concerns have been shown to outweigh the perceived disease risks in determining vaccine acceptance as indicated in a systematic review among health care workers in Africa [36]. Informing the public about the safety of a COVID-19 vaccine should be the focus for health authorities aiming to achieve a high vaccine uptake especially in Tanzania where other factors including contradicting government stance may have had an influential role in overall vaccine acceptancy.

As expected, this study showed high perceived risk for COVID-19 among older HCW which correlated with high vaccine uptake. Literature indicates that vaccine hesitancy was more common among young people than older adults partly due to their lower risk of comorbidities [37–39]. Further, the observed excess mortality in the elderly population due to COVID-19 may have functioned to make this group feel particularly vulnerable, thus both enhancing their risk perception and increasing willingness to adopt protective measures. Being male has been reported to be uniformly associated with lower risk perceptions in many countries, which is consistent with other risk perception studies [40], a finding which was not corroborated in the present study. This may be due to the similarity of our participants with respect to the perceived risk of getting COVID-19 infection. That is, all HCWs have same risk for the infection, irrespective of their identified sex.

The triangulation method used in this study under mixed method design provides a deeper understanding and contextual insights of the research in question. The key informant interviews and the focused group discussions complement the quantitative findings. One major limitation relates to cross-sectional nature of current study as the vaccine was being introduced in Tanzania and following a COVID-19 surge. Attitudes and uptake may change over time as the pandemic continues and roll-out expands. Therefore, our findings should be interpreted with caution with an understanding that the situation is likely to evolve over time. The second limitation is a possibility of recall bias, particularly due to inability of using HCWs electronic COVID-19 vaccination records to confirm vaccination history. However, this limitation is not expected to greatly influence current findings due to the relatively short time between the initial vaccination roll-out and our study. Thirdly, in conducting a mixed method study there is the possibility of introducing interviewer bias. To minimize this, in addition to training, authors provided a common interviewer guide to every interviewer to enhance objectivity. Despite these limitations, findings presented provide valid and important insights into the risk perceptions of the HCWs and associated structural factors during the COVID-19 pandemic in Tanzania.

## Conclusions

While majority of the HCWs perceived to have a high risk of contracting COVID-19, only about a half of respondents reported to be vaccinated. High perceived risk for COVID-19

infection, older age, female gender, working in a district hospital and a privately owned hospital were associated with increased vaccine uptake. The prevailing working environment and constant exposure to patients, rendered some HCWs to perceive their risk of contracting COVID-19 as unavoidable. However, other HCWs declared the potential role of COVID-19 vaccines in reducing their risk of infection. Health promotion activities focusing on the beneficial role of COVID-19 vaccine in reducing transmission risk may increase vaccine uptake in Tanzania.

## Policy implications

A consistent and evidence-based position adopted by the health authorities is an important prerequisite towards addressing any novel public health emergency. Tackling future public health emergencies requires tailored, effective health promotion measures to encourage uptake of protective interventions. HCWs should be the primary targets of health promoting activities since they remain to be key sources of information for the general population and their personal attitudes greatly influence communities' intervention uptake. Such deliberate and targeted health promotion actions are crucial in the context where policies allow for voluntary intervention uptake, as is the case in Tanzania.

## Acknowledgments

The authors acknowledge their corresponding institutions for providing support to conduct the study. We recognize the cooperation from all the health care workers that took part in this study. We acknowledge the support from the regional and district medical and vaccine officers and the health facilities-in charges of the involved health facilities. We thank our research assistants for their dedication to conduct this study timely, namely Melina Mgongo, Doris Mbata, Oko Okong'o, Zenaice Aloyce, Martha Joseph, Mtumwa Bakari, Nyanjura Manyama, Zenais Kiwale, Anastazia Ngowi, Barikiel Panga, Novatus Tesha, Albert Majura, Naike Nathaniel, Julietha Tibyesiga, Loveness Kimaro, Judith Kokuleba, Ngusa Kalambo, Constancia F Luyenga, Monica Mtei, Jackline Ngowi, Edson B Jeremiah, Witness Simon, Erick Kazoka, Chrispin Mgute.

## Author Contributions

**Conceptualization:** Kasusu Klint Nyamuryekung'e, Maryam Amour, Innocent Mboya, Harrieth Ndumwa, James Kengia, Belinda J. Njiro, Lwidiko Mhamilawa, Elizabeth Shayo, Frida Ngalesoni, Ntuli Kapologwe, Emmy Metta, Sia Msuya.

**Formal analysis:** Kasusu Klint Nyamuryekung'e, Maryam Amour, Harrieth Ndumwa, Lwidiko Mhamilawa, Elizabeth Shayo, Frida Ngalesoni, Ntuli Kapologwe, Albino Kalolo, Emmy Metta.

**Funding acquisition:** Maryam Amour, James Kengia, Ntuli Kapologwe, Sia Msuya.

**Investigation:** Kasusu Klint Nyamuryekung'e, Maryam Amour, Innocent Mboya, Harrieth Ndumwa, Belinda J. Njiro, Lwidiko Mhamilawa, Elizabeth Shayo, Frida Ngalesoni, Albino Kalolo, Emmy Metta.

**Methodology:** Kasusu Klint Nyamuryekung'e, Maryam Amour, Innocent Mboya, Harrieth Ndumwa, James Kengia, Belinda J. Njiro, Lwidiko Mhamilawa, Elizabeth Shayo, Frida Ngalesoni, Ntuli Kapologwe, Albino Kalolo, Emmy Metta, Sia Msuya.

**Project administration:** Kasusu Klint Nyamuryekung'e, Harrieth Ndumwa, Belinda J. Njiro, Elizabeth Shayo.

**Resources:** Maryam Amour, James Kengia, Ntuli Kapologwe.

**Software:** Innocent Mboya, Lwidiko Mhamilawa, Albino Kalolo.

**Supervision:** Kasusu Klint Nyamuryekung'e, Maryam Amour, Innocent Mboya, James Kengia, Elizabeth Shayo, Frida Ngalesoni, Ntuli Kapologwe, Albino Kalolo, Emmy Metta, Sia Msuya.

**Validation:** Innocent Mboya, Harrieth Ndumwa, Lwidiko Mhamilawa, Sia Msuya.

**Writing – original draft:** Kasusu Klint Nyamuryekung'e, Frida Ngalesoni, Emmy Metta.

**Writing – review & editing:** Kasusu Klint Nyamuryekung'e, Maryam Amour, Innocent Mboya, Harrieth Ndumwa, James Kengia, Belinda J. Njiro, Lwidiko Mhamilawa, Elizabeth Shayo, Ntuli Kapologwe, Albino Kalolo, Sia Msuya.

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
