## [Decision Letter · Decision Letter 0]

16 Feb 2023

PGPH-D-22-01599

Health care workers’ self-perceived infection risk and COVID-19 vaccine uptake: a mixed methods study

Dear Dr. Nyamuryekung'e,

Thank you for submitting your manuscript to PLOS Global Public Health. After careful consideration, we feel that it has merit but does not fully meet PLOS Global Public Health’s publication criteria as it currently stands. Therefore, we invite you to submit a revised version of the manuscript that addresses the points raised during the review process.

EDITOR: Kindly take careful consideration of the reviewers comments in revising your manuscript. In addition, I have the following comments consistent with those of the reviewers:

As part of the introduction study, the authors should better contextualize the study by addressing the availability of a national policy / guideline on vaccination of health care workers against preventable diseases (e.g. hepatitis B, influenza) including COVID-19. Also provide information on the current COVID-19 vaccine coverage rates within the general population, and among relevant populations groups like health care workers (globally versus regionally versus locally in Tanzania).

The authors should provide a justification for choosing thematic analysis as the most appropriate method for the data collected. Further to this, a full outline / list of the core emergent themes should be explicitly reported in the Results section.

The Discussion section as it currently stands is fairly thin. The authors should fully appraise the findings of this study against similar studies conducted in Africa and elsewhere (providing specific examples where relevant), exploring health care workers’ perceived risk of COVID-19 and uptake of COVID-19 vaccination.

The limitations of this study are not adequately addressed.  For example, the authors should address issues around recall bias and the limitation of not using health care workers’ electronic COVID-19 vaccination records/certificates to confirm vaccination history. It is also unclear to me if health care workers were asked to provide reasons for why they were not vaccinated. Is there a reason the authors chose not to collect such data.

The authors should take note of editorial and grammatical errors throughout the document and ensure that these are resolved when revising the manuscript. Examples include but are not limited to the following:

Line 79 pg 8; “…first nation-wise COVID 19 vaccination…” Do the authors mean “nation-wide”?Inconsistency in writing COVID-19 and SARS-CoV-2Lines 97 – 99 pg 9; “While being a high-risk group…associated positive protective behaviors” This sentence is unclear. Kindly rephrase for better clarity.Line 235 pg 17; “The higher perceived risk to COVID 19 infection perceptions…” Please rephrase this sentence for ease of clarity.Line 325 pg 21; “However, consisted availability of appropriate and required PPEs…” Please rephrase this sentence for ease of clarity.Lines 375 – 377 pg 24; “Targeted information the public on the beneficial role…” Please rephrase for clarity.

Lines 82 – 83 pg 8; “A total of 2,431,769 vaccine doses administered by the end of year 2021 under COVAX facility” this sentence appears to be hanging or incomplete. No date as to when this tally was reach has been supplied.

Lines 83 – 85 pg 8; “While vaccine availability may not have been a challenging option initially…” This statement may be misleading. The challenges in availability of, and access to COVID-19 vaccines across the African region at the height of the pandemic have been widely acknowledged. Further to this, the negative far-reaching consequences of withholding vaccines from the region on vaccine acceptance and uptake at the population level may be immeasurable but existent. Please consider rephrasing this statement.

The sample size in the Abstract section is 1386 while that in the results section is 1368. Please resolve this disparity. In addition, kindly address if all health care workers invited agreed to participate in this study. In other words, what are the recruitment versus enrolment numbers for this study.

While Table 3 is presented, it is not cited within the text. Kindly make reference to Table 3 within the reference section for ease of clarity.

Lines 336 – 337 pg 22; “It has been widely reported that high perceived risk of contracting COVID 19 is a significant predictor of vaccine acceptance.” Please provide reliable references to support this statement.

Line 340 – 342 pg 22; “One probable explanation for this…people in Africa consider the vaccines as unnecessary…” Given that this was a single country, multi-site study, is this inference and generalization founded?

Line 344 – 347 pg 22; “However, this study has shown…especially vaccine safety concerns…” did the authors report on vaccine safety concerns among the study population?

I was unable to access the supplementary file “Data review URC” using the link provided.

We look forward to receiving your revised manuscript.

Kind regards,

Edina Amponsah-Dacosta, Ph.D., MPH

Academic Editor

Journal Requirements:

1. In the online submission form, you indicated that your data will be submitted to the Dryad database upon acceptance. Should your submission be accepted, we will require the following information in your Data Availability Statement: 

a. The DOI provided by Dryad

b. The citation for your data package in the reference section of your manuscript

c. The citation for your data package in the methods section

If you are unable to adhere to our open data policy, please kindly revise your statement to explain your reasoning and we will seek the editor's input on an exemption. Please be assured that, once you have provided your new statement, the assessment of your exemption will not hold up the peer review process.

Additional Editor Comments (if provided):

Reviewers' comments:

Reviewer's Responses to Questions

**Comments to the Author**

1. Does this manuscript meet PLOS Global Public Health’s publication criteria? Is the manuscript technically sound, and do the data support the conclusions? The manuscript must describe methodologically and ethically rigorous research with conclusions that are appropriately drawn based on the data presented.

Reviewer #1: Yes

Reviewer #2: Partly

2. Has the statistical analysis been performed appropriately and rigorously?

Reviewer #1: No

Reviewer #2: No

3. Have the authors made all data underlying the findings in their manuscript fully available (please refer to the Data Availability Statement at the start of the manuscript PDF file)?

Reviewer #1: Yes

Reviewer #2: No

4. Is the manuscript presented in an intelligible fashion and written in standard English?

Reviewer #1: Yes

Reviewer #2: Yes

5. Review Comments to the Author

Reviewer #1: Firstly I would like to congratulate the authors on the equality of their manuscript.

I would like to recommend if they consider using Univariate binary logistic regression to estimate the magnitude of the statistical associations and perform multivariate logistic regression to identify the predictors of SARS-CoV-2 vaccine uptake and high risk perception. In addition to controlling for potential confounders. Findings from the current study would have stronger evidence if both descriptive and analytical analysis. If vaccination dates are available analyzing risk perception per pandemic situation period would be valuable as well. In the discussion part I would recommend mentioning changes in pandemic situation between November 2021 and January 2022 which could influence COVID-19 risk perception among HWCs.

Reviewer #2: Congratulations on conducting important paper regarding Health care workers’ self-perceived infection risk and COVID-19 vaccine uptake.

The manuscript is logical and of a reasonable standard, being based on sound scientific principles. However, there are several issues (highlighted below) which have raised my concern.

1. The manuscript does not read well in some sections with many instances of poor grammar. I strongly recommend that you make use of a language editor.

2. The abstract contains abbreviations/acronyms which have not been previously defined.

3. The authors mix spellings of American English and UK English. I would suggest they stick to one style throughout

4. The data of confirmed COVID-19 cases and deaths is very dated. The same for number of vaccine doses administered under the COVAX facility. Please update with latest data

5. Acceptability, and willingness of the population to vaccinate is tautology.

6. Ensure consistency in terminology; COVID-19 versus COVID 19. SARS-CoV-2

7. Line 79, nation-wide instead of nation-wise?

8. Please add a reference to the bold statement in lines 83 – 85.

9. Add quotation marks to the definition of vaccine hesitancy (lines 104 -105).

10. Study sites – It is not clear the reason why Dar es salaam was purposefully chosen?

11. Sample size determination – what is the population from which the sample size determination (line 141-143) was determined?

12. How were HCWs chosen for participation in the study? Were all available HCWs invited to participate and interviews carried out until the minimum sample size was reached? Or were HCWs randomized for participation in the study? This is not clear.

13. HCWs was defined earlier in the manuscript. No need to use the full term thereafter.

14. Please add details about the pilot test.

15. Is there any special reason why the qualitative component of the study was not carried out in all seven regions? Why the choice of four regions instead of all regions (or the one cosmopolitan (Dar es salaam) region)?

16. Is ‘hospital in charges’ (line 157) an official designation or title in the Tanzanian health system?

17. More details about the questionnaire. What did the validated tool contain? Likert scale to measure risk perception?

18. Line 170 - ‘Consent information was administered?’ Do you mean the study information was given and consent obtained?

19. Line 175, please define the abbreviations, IDI and FDG in full.

20. It is not clear how the authors determined high risk perception and low risk perception. What was the cut-off point at which a participant can be regarded as having a high risk perception to COVID-19?

21. RESULTS – Please state the response rate of the participants.

22. RESULTS – Table 1, Intervals for the work experience (<6 and 6+) are not logical.

23. Table 2 results should be shown as Odds Ratio. This would be a more a useful result.

24. Please use the same number of decimal places for your percentages

25. More probing of participants of the qualitative arm of the study in order to document their risk behaviours, experiences, and even recommendations on how to improve the vaccine uptake and/or risk perception.

26. The Discussion is very thin. The authors missed out on an opportunity to discuss their findings and compare with many other studies carried out in Tanzania (https://www.mdpi.com/2076-393X/10/9/1429;
https://iariw.org/wp-content/uploads/2022/10/Masele_Daud_IARIW-TNBS-2022-1.pdf) and other international studies such as (https://journals.plos.org/plosone/article?id=10.1371/journal.pone.0250017;
https://www.frontiersin.org/articles/10.3389/fpsyg.2020.02166/full;
https://journals.plos.org/plosone/article?id=10.1371/journal.pone.0242471;
https://www.tandfonline.com/doi/full/10.2147/RMHP.S310289;
https://intjem.biomedcentral.com/articles/10.1186/s12245-021-00341-0;
https://www.sciencedirect.com/science/article/pii/S0195670120302784?casa_token=OvEn9vM2IzQAAAAA:umHRhFRZiScbi163sbEXlt0r6gV1ZFMAJcPpsPu979DDzqi4nhdqqjwCSH1TOEleXf2kanJp4wPg) to mention a few studies.

27. Clear recommendations should be provided based on the findings of the study.

28. The statements under the ‘Policy implications’ section of the manuscript are too general. The authors should state how future/current policy for vaccination and risk behaviours can be influenced and by the findings of the study

29. Reference #3 and #6 needs to be fixed (start with the author).

30. Add access dates to all online references

6. PLOS authors have the option to publish the peer review history of their article (what does this mean?). If published, this will include your full peer review and any attached files.

**Do you want your identity to be public for this peer review?** For information about this choice, including consent withdrawal, please see our Privacy Policy.

Reviewer #1: No

Reviewer #2: No

---

## [Decision Letter · Decision Letter 1]

26 Apr 2023

PGPH-D-22-01599R1

Health care workers’ self-perceived infection risk and COVID-19 vaccine uptake: a mixed methods study

Dear Dr. Nyamuryekung'e,

Thank you for submitting your revised manuscript to PLOS Global Public Health. After careful consideration, the reviewers feel that there are still some aspects of the manuscript that still need revision. Therefore, we invite you to submit a revised version of the manuscript that addresses the points raised during the review process. Please take the time to carefully address these minor comments as this will be your final opportunity to submit a revised draft for consideration.

We look forward to receiving your revised manuscript.

Kind regards,

Edina Amponsah-Dacosta, Ph.D., MPH

Academic Editor

Journal Requirements:

Additional Editor Comments (if provided):

Reviewers' comments:

Reviewer's Responses to Questions

**Comments to the Author**

1. If the authors have adequately addressed your comments raised in a previous round of review and you feel that this manuscript is now acceptable for publication, you may indicate that here to bypass the “Comments to the Author” section, enter your conflict of interest statement in the “Confidential to Editor” section, and submit your "Accept" recommendation.

Reviewer #1: All comments have been addressed

Reviewer #2: (No Response)

2. Does this manuscript meet PLOS Global Public Health’s publication criteria? Is the manuscript technically sound, and do the data support the conclusions? The manuscript must describe methodologically and ethically rigorous research with conclusions that are appropriately drawn based on the data presented.

Reviewer #1: Yes

Reviewer #2: Yes

3. Has the statistical analysis been performed appropriately and rigorously?

Reviewer #1: Yes

Reviewer #2: Yes

4. Have the authors made all data underlying the findings in their manuscript fully available (please refer to the Data Availability Statement at the start of the manuscript PDF file)?

Reviewer #1: Yes

Reviewer #2: Yes

5. Is the manuscript presented in an intelligible fashion and written in standard English?

Reviewer #1: Yes

Reviewer #2: Yes

6. Review Comments to the Author

Reviewer #1: I thank the authors for addressing all my comments and made an adequate revision of the submitted article. I have no further comment with regards to the manuscript.

Reviewer #2: A few remarks remain unresolved. Please address the following:

1. Line 3 of the abstract; ‘healthcare’ is different for that used in the title ‘health care’. Also check Lines 80, 143, 244, 299.

2. HCWs is defined in line 80. No need to use the unabbreviated text thereafter.

3. Are the terms healthcare workers, healthcare professionals, healthcare personnel interchangeable? I recommend using health care workers throughout the manuscript.

4. Line 68-69. The data of confirmed global COVID-19 cases and deaths is very dated. Please update with latest data

5. Line 192, FGD already defined earlier.

6. Line 265, change COVID 19 to COVID-19

7. Line 299, HCWs instead of health care workers.

8. Table 2 – please double check the 95% CI values. Does not seem correct

9. Reference #2 is incomplete.

10. The authors in their response claim the following ‘A more extensive appraisal of current findings against available relevant literature has now been conducted. Several papers have been added in the related to similar papers in Africa and from recommendations provided by the reviewers.’ This is not reflected in the discussion with the same references used as in the original submission.

7. PLOS authors have the option to publish the peer review history of their article (what does this mean?). If published, this will include your full peer review and any attached files.

**Do you want your identity to be public for this peer review?** For information about this choice, including consent withdrawal, please see our Privacy Policy.

Reviewer #1: No

Reviewer #2: **Yes: **Mncengeli Sibanda

---

## [Decision Letter · Decision Letter 2]

8 May 2023

Health care workers’ self-perceived infection risk and COVID-19 vaccine uptake: a mixed methods study

PGPH-D-22-01599R2

Dear Dr. Nyamuryekung'e,

We are pleased to inform you that your manuscript 'Health care workers’ self-perceived infection risk and COVID-19 vaccine uptake: a mixed methods study' has been provisionally accepted for publication in PLOS Global Public Health.

Best regards,

Edina Amponsah-Dacosta, Ph.D., MPH

Academic Editor

Reviewer Comments (if any, and for reference):

Reviewer's Responses to Questions

**Comments to the Author**

1. If the authors have adequately addressed your comments raised in a previous round of review and you feel that this manuscript is now acceptable for publication, you may indicate that here to bypass the “Comments to the Author” section, enter your conflict of interest statement in the “Confidential to Editor” section, and submit your "Accept" recommendation.

Reviewer #2: All comments have been addressed

2. Does this manuscript meet PLOS Global Public Health’s publication criteria? Is the manuscript technically sound, and do the data support the conclusions? The manuscript must describe methodologically and ethically rigorous research with conclusions that are appropriately drawn based on the data presented.

Reviewer #2: Yes

3. Has the statistical analysis been performed appropriately and rigorously?

Reviewer #2: Yes

4. Have the authors made all data underlying the findings in their manuscript fully available (please refer to the Data Availability Statement at the start of the manuscript PDF file)?

Reviewer #2: Yes

5. Is the manuscript presented in an intelligible fashion and written in standard English?

Reviewer #2: Yes

6. Review Comments to the Author

Reviewer #2: (No Response)

7. PLOS authors have the option to publish the peer review history of their article (what does this mean?). If published, this will include your full peer review and any attached files.

**Do you want your identity to be public for this peer review?** For information about this choice, including consent withdrawal, please see our Privacy Policy.

Reviewer #2: No
